# iNOS Deletion in Alveolar Epithelium Cannot Reverse the Elastase-Induced Emphysema in Mice

**DOI:** 10.3390/cells12010125

**Published:** 2022-12-28

**Authors:** Marija Gredic, Vinita Sharma, Stefan Hadzic, Cheng-Yu Wu, Oleg Pak, Baktybek Kojonazarov, Julia Duerr, Marcus A. Mall, Andreas Guenther, Ralph T. Schermuly, Friedrich Grimminger, Werner Seeger, Simone Kraut, Natascha Sommer, Norbert Weissmann

**Affiliations:** 1Cardio-Pulmonary Institute (CPI), Universities of Giessen and Marburg Lung Center (UGMLC), Member of the German Center for Lung Research (DZL), Justus-Liebig-University, 35392 Giessen, Germany; 2Institute for Lung Health (ILH), Justus-Liebig-University, 35392 Giessen, Germany; 3Department of Translational Pulmonology, University of Heidelberg, 69117 Heidelberg, Germany; 4Translational Lung Research Center (TLRC), German Center for Lung Research (DZL), 69120 Heidelberg, Germany; 5Department of Pediatric Respiratory Medicine, Immunology and Critical Care Medicine, Charité-Universitätsmedizin Berlin, Corporate Member of Freie Universität Berlin and Humboldt-Universität zu Berlin, 13353 Berlin, Germany; 6Berlin Institute of Health at Charité, Universitätsmedizin Berlin, Charitéplatz 1, 10117 Berlin, Germany; 7European IPF Registry & Biobank (eurIPFreg), 35392 Giessen, Germany; 8Agaplesion Evangelisches Krankenhaus Mittelhessen, 35398 Giessen, Germany; 9Max Planck Institute for Heart and Lung Research, Member of the German Center for Lung Research (DZL), 61231 Bad Nauheim, Germany

**Keywords:** COPD, emphysema, iNOS, lung epithelium, AECII

## Abstract

Background: Chronic obstructive pulmonary disease (COPD) is the third leading cause of death worldwide. In addition to chronic bronchitis and emphysema, patients often develop at least mild pulmonary hypertension (PH). We previously demonstrated that inhibition of inducible nitric oxide synthase (iNOS) prevents and reverses emphysema and PH in mice. Interestingly, strong iNOS upregulation was found in alveolar epithelial type II cells (AECII) in emphysematous murine lungs, and peroxynitrite, which can be formed from iNOS-derived NO, was shown to induce AECII apoptosis in vitro. However, the specific cell type(s) that drive(s) iNOS-dependent lung regeneration in emphysema/PH has (have) not been identified yet. Aim: we tested whether iNOS knockout in AECII affects established elastase-induced emphysema in mice. Methods: four weeks after a single intratracheal instillation of porcine pancreatic elastase for the induction of emphysema and PH, we induced iNOS knockout in AECII in mice, and gave an additional twelve weeks for the potential recovery. Results: iNOS knockout in AECII did not reduce elastase-induced functional and structural lung changes such as increased lung compliance, decreased mean linear intercept and increased airspace, decreased right ventricular function, increased right ventricular systolic pressure and increased pulmonary vascular muscularization. In vitro, iNOS inhibition did not reduce apoptosis of AECII following exposure to a noxious stimulus. Conclusion: taken together, our data demonstrate that iNOS deletion in AECII is not sufficient for the regeneration of emphysematous murine lungs, and suggest that iNOS expression in pulmonary vascular or stromal cells might be critically important in this regard.

## 1. Introduction

Chronic obstructive pulmonary disease (COPD), a heterogeneous condition that encompasses chronic bronchitis and emphysema, is the third leading cause of death worldwide [1,2]. The disease is progressive, and currently incurable, with the treatment options limited to those controlling symptoms. The main cause of COPD is tobacco smoking, or chronic exposure to some other noxious gases, in complex interaction with genetic and developmental factors. In addition, single-gene defect (α1-anti-trypsin deficiency), severe asthma, infections, abnormal lung development and/or failed lung regeneration can cause or contribute to COPD development [3,4,5,6]. The underlying molecular mechanisms of COPD pathology include increased oxidative and nitrosative stress, an imbalance between proteolytic activity and anti-proteolytic defense, persistent inflammation in the lung, uncontrolled autophagy, enhanced apoptosis and/or accelerated lung aging [4,7]. In addition to respiratory symptoms, most COPD patients suffer from mild to moderate pulmonary hypertension (PH) [8,9]. We have previously described the essential role of the inducible nitric oxide synthase (iNOS) in the development and reversal of tobacco smoke-induced emphysema and PH in mice [10]. More recently, we demonstrated that iNOS inhibition ameliorates parenchymal destruction and promotes reverse remodeling of the pulmonary vasculature, even in a severe model of elastase-induced emphysema, characterized by prominent parenchymal damage similar to the lesions found in lungs of end-stage COPD patients [11]. Regarding the molecular mechanism, evidence was provided that protein nitration, prompted by the formation of peroxynitrite from iNOS-derived nitric oxide (NO) and superoxide produced by the NADPH oxidase comprising the NADPH oxidase organizer (NoxO)-1 subunit, may drive lung emphysema development [10,12]. Interestingly, while it was clear that both vascular and alveolar changes occur in an iNOS-dependent manner in mice, we observed that they are associated with iNOS deregulation in different cell types. In particular, we showed that the presence of iNOS in myeloid cells leads to pulmonary vascular remodeling but does not contribute to emphysema development [10,13]. However, it remained unclear which pulmonary vascular, stromal, or epithelial cell type was the main source of the iNOS activity linked to pathological changes in the lung parenchyma. In this regard, Boyer and colleagues published that alveolar epithelial type II cells (AECII) are characterized by marked iNOS upregulation in the elastase model of emphysema and by a more prominent accumulation of nitrated proteins than any other lung cell type [14]. While the authors did not observe any beneficial effect of preventive iNOS inhibition on the emphysema development monitored up to 20 days after elastase instillation, the abovementioned recent work from our group showed that a longer 12-week treatment with an iNOS inhibitor can ameliorate fully established emphysema and PH in this animal model [11]. This finding suggests that iNOS inhibition may be a viable option for the treatment of severe emphysema in mice if present during a longer period of time. However, it was unresolved which cell type was driving lung regeneration upon iNOS inhibition in this model. As the delineation of the exact signaling pathway through which iNOS exerts its pathological effects is of vast importance for potential future transfer to the human disease, and might impact the route of administration, we decided to further investigate which cell type is behind pathological iNOS activity and the reversal of this disease. Previously, strong evidence was provided that AECll have stem-cell features in the adult lung, and proliferate to repopulate and repair the injured epithelium [15,16]. We thus hypothesized that increased iNOS activity, the subsequent formation of peroxynitrite, and the accumulation of nitrated proteins in AECII cells could impair their potential for self-renewal, and disable the regeneration of respiratory epithelium. Therefore, this study aimed to investigate whether the deletion of iNOS in the AECII can promote the reversal of established severe emphysema if the knockout is present during the 12 weeks post-injury in the elastase mouse model. 

## 2. Materials and Methods

### 2.1. Experimental Design and Elastase Application

The mice were housed in individually ventilated cages, with a 12-h dark/light cycle and food and water supplied ad libitum. Male and female *iNos CCSP rtTA2^s^-M2 LC1* (B6. Nos2^tm2904.1Arte^ C(Cg)-Tg(Scgb1a1-rtTA)38-Tg(tetO-cre)LC1Bjd) (Figure 1B), 3–4 months old, were randomly allocated to four experimental groups: saline-instilled control mice, saline-instilled doxycycline-treated mice, porcine pancreatic elastase (PPE)-instilled control or PPE-instilled doxycycline-treated mice. First, 100 µL saline or elastase (E7885; Merck KGaA, Darmstadt, Germany) dissolved in saline (24 U/kg body weight) was delivered intratracheally to the anesthetized animals (3% isoflurane in 100% O_2_) using a MicroSprayer^®^ Aerosolizer (Model IA-1C; Penn-Century, Inc., Wyndmoor, PA, USA) with a high-pressure syringe (Model FMJ-250; Penn-Century, Inc., Wyndmoor, PA, USA). Then, from the 14th to the 28th day post instillation, mice from the appropriate experimental groups were fed with chow containing doxycycline (600 mg/kg supplemented with 2% saccharose; Altromin Spezialfutter GmbH & Co., Lage, Germany). After this point, the mice were given an additional 12 weeks for possible recovery, prior to the in vivo measurements: µCT, echocardiography, hemodynamic, and lung function measurements (Figure 1F). The final numbers in these measurements may vary because of technical problems, e.g., with the placement of the hemodynamic catheter or the echocardiographic assessment. All analyses of the in vivo experiments were performed at the end of the observation period, and the histological and molecular biology analyses were performed using material thereof. For confirmation of iNOS knockout induction in AECII by doxycycline, lungs were harvested from the mice immediately or 12 weeks after termination of feeding with doxycycline-containing chow. Isolated AECII from these lungs or whole-lung sections were investigated for luciferase and iNOS expression, respectively. All animal experiments were approved by the regional authorities for animal welfare (Regierungspräsidium Giessen, Germany), in accordance with the German animal welfare law and the European legislation for the protection of animals used for scientific purposes (2010/63/EU).

### 2.2. µCT and Echocardiography 

Micro-computed tomography (µCT) and echocardiography were carried out under isoflurane (Baxter Deutschland GmbH, Unterschleiβheim, Germany) anesthesia by a blinded single observer, as described previously [17]. The following parameters were measured: functional residual capacity (FRC) and lung density using µCT, and pulmonary acceleration time, pulmonary ejection time, tricuspid annular plane systolic excursion, and right ventricular (RV) wall thickness using echocardiography.

### 2.3. Bronchoalveolar Lavage and Lung Tissue Processing

After the hemodynamic measurement, the thoracic cavity was opened, and bronchoalveolar lavage (BAL) was carried out three times, using 700 µL ice-cold PBS buffer. The collected BAL was then centrifuged for 10 min at 4 °C at a speed of 2490× *g* (Micro 200R, Hettich, Germany), and separated into BAL fluid (BALF), stored at −80 °C) and BAL cells (resuspended in FBS with 10% DMSO and kept at −80 °C). The blood was flushed out of the lungs through the pulmonary artery with the saline solution, at a pressure of 22 cmH_2_O during continuous ventilation (Minivent, Hugo Sachs Electronik, March, Germany). The right lung was harvested for molecular biology analyses, while the left lung was fixed under simultaneous vascular perfusion (pressure: 22 cmH_2_O) and inflation (pressure: 12 cmH_2_O) with formalin.

### 2.4. Lung Function and Hemodynamic Measurements

Lung function measurement and hemodynamic measurements were carried out under isoflurane anesthesia, as previously reported. Deep inflation was used as a recruitment maneuver, followed by single-frequency forced oscillation and broadband forced oscillation, and the measurement of a respiratory pressure–volume (P–V) loop [13,17]. After lung function assessment, a micro-tip catheter (Millar Instruments, Houston, TX, USA) was placed in the right ventricle through the surgically prepared jugular vein for measurement of the right ventricular systolic pressure (RVSP). The animals were sacrificed under anesthesia by exsanguination through the carotid artery.

### 2.5. Right-Heart-Hypertrophy Assessment

After lung fixation with formalin, the heart was removed, dissected into the RV and the left ventricle plus septum (LV + S), and weighed. The Fulton index (RV/(LV + S)) was calculated as a measure of RV hypertrophy.

### 2.6. Alveolar Morphometry 

For alveolar morphometry, formalin-fixed and paraffin-embedded 3 µm thick lung sections were stained with hematoxylin and eosin (H&E). The sections were analyzed using the Qwin alveolar morphometry software, as previously described [10,11,13], to assess alveolar wall thickness, airspace, and mean linear intercept. The bronchi and vessels were excluded from the analysis.

### 2.7. Isolation of Murine Primary AECII and Metabolic Activity Assay

Primary AECII were isolated from the lungs of mice fed with either standard or doxycycline-containing chow, in accordance with the previously described protocol [12]. The cells were either immediately lysed and proteins extracted, or they were seeded in 96-well plates at density 35,000 cells/well in DMEM (high glucose, Gibco, Thermo Fisher Scientific Inc., Waltham, MA, USA), supplemented with 10% (*v/v*) fetal bovine serum (FBS, Sigma Aldrich, Munich, Germany), 2% (*v/v*) L-Glutamine, 100 IU/mL penicillin, and 100 µg/mL streptomycin (all from Gibco, Thermo Fisher Scientific Inc., Waltham, MA, USA), and left overnight in a humidified atmosphere of 5% CO_2_, at 37 °C. The cells were treated with different concentrations of cigarette smoke extract (CSE) with or without 10 µM (L-NIL) for 6 h, and then the medium was replaced with fresh medium containing 10% (*v/v*) alamarBlue Cell Viability Reagent (Thermo Fisher Scientific Inc., Waltham, MA, USA). Metabolic activity was measured 24 h after the start of CSE treatment.

### 2.8. Metabolic Activity Assay

The MLE 12 (ATCC CRL-2110) cells were seeded in 96 well plates at 10,000 cells/well in DMEM/F-12 (Gibco, Thermo Fisher Scientific Inc., Waltham, MA, USA), supplemented with 10% (*v*/*v*) FBS (Sigma Aldrich, Munich, Germany), 100 IU/mL penicillin, and 100 µg/mL streptomycin (both from Gibco, Thermo Fisher Scientific Inc., Waltham, MA, USA), and cultured overnight in a humidified atmosphere of 5% CO_2_, at 37 °C. The cells were then serum-starved for 24 h. For the experiments with CSE, the next day, the cells were treated with different CSE concentrations with or without 10 µM L-NIL for 6 h, and then kept for the additional 16 h after the treatment removal and addition of a fresh culture medium. For experiments with BALF, the cells were treated with BALF diluted 1:10 in a serum-free medium for 24 h. PBS served as a control. The experiment was repeated 18 times in total, by treating the cells in three different experiments with BALF from six different animals per group. Afterwards, the BALF treatment was removed and a new culture medium with 10% (*v/v*) alamarBlue Cell Viability Reagent (Thermo Fisher Scientific Inc., Waltham, MA, USA) was added. The metabolic activity of the cells was determined after 4 h, in accordance with the manufacturer’s instructions. 

### 2.9. Cell Proliferation Assay

The cells were seeded and serum-starved, as described for the metabolic activity assay. Then, the cells were treated with different concentrations of CSE with or without 10 µM L-NIL, for 6 h. The medium was replaced with a fresh medium containing BrdU labeling solution (Roche, Basel, Switzerland), and the cells were cultured for an additional 16 h. For the experiments with BALF, a serum-free medium containing BrdU labeling solution (Roche, Basel, Switzerland) and either BALF diluted 1:10 or PBS (as a control) was added, and the cells were cultured for 16 h. Next, the cells were fixed and proliferation was determined, in accordance with the manufacturer’s instructions. The absorbance was measured in a microplate reader at 370 nm (reference wavelength: 492 nm). The O.D. value for each experimental sample was standardized to the value measured in the control group.

### 2.10. Cell Apoptosis Assay

The cells were seeded in 96-well plates at 10,000 cells/well and left overnight in a humidified atmosphere of 5% CO_2_, at 37 °C. The next day, the cells were treated with different concentrations of CSE with or without 10 µM L-NIL for 6 h, and then cultured in a new, fresh medium with or without L-NIL, in the presence of propidium iodide and Annexin XII-based polarity-sensitive probe pSIVA^TM^ from the apoptosis kinetic assay (Abcam, Cambridge, UK), for an additional 16 h. For the BALF experiments, fresh serum-free medium containing either BALF diluted 1:10 or PBS was added, together with propidium iodide and Annexin XII-based polarity-sensitive probe pSIVA^TM^ (Abcam, Cambridge, UK). The assay was carried out in accordance with the manufacturer’s instructions, using an IncuCyte ZOOM device (Essen BioScience, Ann Arbor, MI, USA) for visualization.

### 2.11. Western-Blot

The isolation of proteins from mouse lung homogenates, protein concentration measurement, and Western-blot analysis were carried out as previously reported [11,13]. The antibodies used were as follows: anti-MMP8 (Cat#78423), anti-cyclin D1 (Cat#134175) anti-cytochrome c (Cat#90529), anti-Bcl2 (Cat#182858) from Abcam, Cambridge, UK, and Bax (Cat#2772S9, Cell Signaling Technology, Danvers, MA, USA), all at a dilution of 1:1000.

### 2.12. Immunofluorescent Staining of Human and Mouse Lung Sections

The immunofluorescent staining of mouse lung sections was carried out as described previously, with slight modifications [13]. Briefly, 3 µm thick sections of formalin-fixed, paraffin-embedded left lung lobes were deparaffinized and rehydrated, and heat-mediated antigen retrieval was carried out using Rodent Decloaker (Biocare Medical, Pacheco, USA). Blocking of non-specific antibody binding was carried out using 10% BSA for 1 h at RT and Rodent M Blocker (Biocare Medical, Pacheco, USA). The sections were incubated overnight at 4 °C with the following primary antibodies: anti-iNOS (Cat#3523, Abcam, Cambridge, UK, 1:200), and anti-vWF (Cat#a0082, Dako, Hamburg, Germany, 1:500). The next day, the slides were washed with 1XPBS and incubated for 2 h at RT with the following fluorescently labeled primary and/or secondary antibodies: anti-Cy3-labeled α-smooth muscle actin antibody (Cat#C6198, Sigma-Aldrich, Munich, Germany 1:400), Alexa fluor488-labeled goat anti-rabbit secondary antibody (Cat#A-11034, Thermo Fisher Scientific Inc., Waltham, MA, USA, 1:400), Zenon™ Rabbit IgG Labeling Kit Alexa fluor 647 (Cat#Z25308, Thermo Fisher Scientific Inc., Waltham, MA, USA)-labeled anti-pro-surfactant protein C (SPC), (Cat#ab211326, Abcam, Cambridge, UK, 1:200). Finally, the slides were counterstained with Hoechst, mounted using ProLong™ Glass Antifade Mountant (Cat#P36980, Thermo Fisher Scientific Inc., Waltham, MA, USA), with precision cover-glasses thickness No. 1.5H, and analyzed using confocal microscopy. At least three images from randomly chosen fields were taken for each analysis. For the human lungs, heat-mediated antigen retrieval was carried out using citrate buffer (Zytomed Systems, Berlin, Germany) and blocking of unspecific binding was carried out using 10% BSA and Background Punisher (Biocare Medical, Pacheco, CA, USA). The following primary antibodies were used: anti-iNOS (Cat#NB300-605, Novus Biologicals, Littleton, CO, USA, 1:250) and anti-HTII280 (Cat# TB-27AHT2-280, Terrace Biotech, San Francisco, CA, USA, 1:200).

### 2.13. Automated Western-Blot Analysis

Automated Western-blot analysis was carried out using the chemiluminescent detection system of the Jess Simple Western™ platform (ProteinSimple, San Jose, CA, USA), in accordance with the manufacturer’s instructions, using an anti-luciferase antibody (Cat#ab21176, Abcam, Cambridge, UK, 1:40).

### 2.14. Statistical Analysis

Statistical analyses were performed using the GraphPad Prism 9 (LaJolla, CA, USA). All data are expressed as means ± standard error of the mean (SEM). A comparison between multiple groups was performed using analysis of variance (a one- or two-way ANOVA for comparing the results of experiments with one or two independent variables, respectively, and Tukey’s multiple comparison post-hoc test). An independent *t*-test was used for comparing the equality of means between the two groups, and *p*-values of < 0.05 were considered statistically significant.

## 3. Results

### 3.1. AECII-Specific Inducible iNOS Deletion Is Achieved Using iNos CCSP rtTA2s-M2 LC1 Line

Considering previous findings showing that iNOS expression and the consequent protein nitration are elevated in AECII in elastase-treated mice, and that treatment with the iNOS inhibitor L-NIL can ameliorate parenchymal damage in this animal model, we wanted to test whether detrimental iNOS activity in emphysema can be attributed to AECII. First, we confirmed the possible relevance for human COPD by demonstrating that iNOS is expressed in AECII in COPD lungs (Figure 1A). Although there was a slight tendency towards an increased intensity of AECII-derived iNOS signal in COPD lungs compared to healthy controls, a prominent upregulation could not be observed. Nevertheless, the possibility remained that the knockout of the protein may facilitate the reversal of the disease, even in the absence of a prominent iNOS upregulation. In this regard, it has been recently shown that the pathogenic effects of iNOS and its role in lung regeneration may depend not only on the amount of the protein in the cell, but also on the context and co-appearance with other proteins and factors such as oxidative stress [12]. Against this background, we next sought to investigate whether iNOS knockout in AECII in mice with fully developed emphysema can promote repair of the respiratory surface. To address this question, we generated AECII-specific iNOS knockout mice, by crossing *iNos^flox/flox^* mice with the recently established *CCSP rtTA2^s^-M2 LC1* line, which enables tight temporal and spatial control of the expression of luciferase and Cre recombinase by doxycycline (Figure 1B) [18]. The successful generation of the mouse line and induction of the knockout was confirmed by quantification of the luciferase expression in isolated primary AECII, and by immunofluorescent analysis of iNOS expression in lungs from the *iNosCCSP rtTA2^s^-M2 LC1* animals fed with standard chow (hereafter controls) and mice with the same genotype fed with doxycycline-containing food (Figure 1C–E). Induction of the knockout in respective experimental groups using doxycycline-containing chow started two weeks after treatment with porcine pancreatic elastase, and finished four weeks after instillation. At this time-point, the emphysema phenotype was previously shown by our laboratory to be fully established and stable in this animal model [11]. Subsequently, animals were given a 12-week period for potential regeneration of parenchyma and reverse remodeling of the vasculature (Figure 1F).

### 3.2. iNOS Deregulation in AECII Does Not Influence the Reversal of PH in the Elastase Model of Emphysema

Hemodynamic measurements 12 weeks after knockout induction showed that RVSP values in elastase-treated animals were the same for control and doxycycline-treated mice, and elevated compared to the saline groups (Figure 2A,B). In addition, echocardiographic measurements showed a decrease in the PAT/PET ratio in elastase-treated animals, independently of iNOS knockout (Figure 2C). Together, these data suggest that iNOS knockout in AECII cannot ameliorate PH in this mouse model. A decline in RV function occurred both in controls and in AECII-specific iNOS knockout mice upon elastase instillation, as shown with echocardiography (Figure 2D). Similarly, echocardiographic assessment of RV wall thickness (RVWT), as well as the Fulton index (Figure 2E,F) demonstrated that the same degree of RV hypertrophy exists in both groups of elastase-treated mice. Finally, muscularization of the pulmonary vasculature, visualized by immunofluorescent staining for α-smooth muscle actin and von Willebrand factor, was increased in the lungs of elastase-treated animals from both groups, compared to saline treatment (Figure 2G), indicating that the AECII-specific iNOS knockout could not promote reverse remodeling of the pulmonary vasculature.

### 3.3. iNOS Knockout in AECII Cells Cannot Improve Lung Function and Does Not Promote Lung Repair in the Elastase Model of Emphysema

In parallel with the investigations into pulmonary vascular changes, we performed in vivo assessment of lung morphology and function in these animals, to evaluate the therapeutic effects of AECII-specific iNOS knockout on emphysema. Functional residual capacity was increased, and lung density decreased to the same level (Figure 3A,B) both in control and doxycycline-treated animals that received elastase, compared to saline groups, suggesting a similar degree of lung damage. Similarly, measurements using an automated FlexiVent system revealed comparable changes in lung function that were consistent with emphysema in elastase-treated animals, irrespective of iNOS knockout in AECII (Figure 3C–E). Curiously, there was a tendency toward a smaller area enclosed by a respiratory pressure-volume loop in elastase-instilled doxycycline-fed mice, compared to controls, which might indicate a reduced amount of atelectasis in these mice (Figure 3F). Finally, histological assessment using alveolar morphometry confirmed that 12 weeks after iNOS knockout induction in AECII, emphysematous changes in the lungs remained the same as those observed in the control animals (Figure 4A–D).

### 3.4. iNOS Activity Does Not Influence Apoptosis or Proliferation of Alveolar Epithelium after Injury In Vivo or In Vitro

Next, we investigated whether the AECII-specific iNOS knockout affects the proliferation and apoptosis rates in the injured epithelium, which were not detectable through our lung function or morphometric analyses, but could be beneficial in the long term for emphysematous lungs. Western-blot analyses demonstrated that levels of the proliferation marker cyclin D1 and the apoptosis marker cleaved caspase 3 in lung homogenates, were unchanged between the control and doxycycline-treated elastase groups (Figure 5A,B). Of note, we detected a baseline difference in cyclin D1 expression between the groups. Interestingly, cytochrome c was downregulated in the lung homogenates of the control, and to a lesser extent, doxycycline-fed elastase-treated mice (Figure 5C). Nevertheless, this change was not accompanied by the differential regulation of Bcl-2 and Bax (Figure 5D,E), nor by the alterations in cleaved caspase 3 content, which suggested that it was limited to the mitochondrial compartment. Interestingly, of all the tested matrix metalloproteinases (MMP-8,-9,-12, data not shown for MMP-9 and -12), only increased MMP-8 expression could be observed in the lungs of elastase-treated animals from both groups, even 16 weeks after the initial injury (Figure 5F).

Furthermore, we tested whether iNOS inhibition in the murine alveolar epithelial cell-line MLE-12 and in murine primary AECII affects their response to a noxious stimulus, in terms of viability, proliferation, and apoptosis. Since, to the best of our knowledge, elastase injury has not been successfully modeled in vitro, and cigarette smoking is the most common and best-known cause of emphysema, we used cigarette smoke extract (CSE) as a toxic challenge for our in vitro experiments. Functional assays on the MLE-12 cell-line (metabolic activity, proliferation, apoptosis; Figure 6A–C) and primary murine AECII (metabolic activity, Figure 6D) demonstrated that iNOS inhibition in these cells does not significantly alter their reaction to CSE. Finally, we treated MLE-12 cells with bronchoalveolar lavage fluid (BALF) from elastase-treated animals, and quantified their proliferation, apoptosis, and metabolic activity (Figure 6E–H). Again, no difference was observed in the effects of BALF taken from control and doxycycline-fed mice. Interestingly, BALF taken from elastase-treated mice inhibited apoptosis, compared to both control (PBS) treatment and treatment with BALF from saline-treated lungs, suggesting that damage control mechanisms mediated by as yet unknown soluble factors take place in lungs injured by elastase. Increased protein concentration in BALF from elastase-treated mice of both genotypes corroborated the suggestion that one or more secreted molecules, proteins or peptides in nature, mediate the observed anti-apoptotic effect of BALF in the MLE-12 cell line (Appendix A).

## 4. Discussion

We investigated whether the induction of iNOS knockout in AECII can promote lung repair or ameliorate pulmonary vascular pathology in mice with fully established elastase-induced emphysema and PH. Our in vivo measurements, performed at the end of a 12-week observation period, revealed statistically significant impairment of heart and lung function and the existence of PH in all elastase-treated mice, irrespective of the iNOS deletion in AECII. These results were corroborated by alveolar morphometry, which showed a prominent, statistically significant development of emphysema in elastase-treated mice that could not be counteracted by iNOS knockout in AECII. Similarly, histological analysis of the small pulmonary vessels indicated that their increased muscularization upon elastase treatment of the lung cannot be reversed by deletion of AECII-derived iNOS. Together, these data demonstrated that iNOS knockout in AECII upon elastase injury in mice does not promote regeneration of alveolar epithelium and reverse remodeling of the pulmonary vasculature. This conclusion was further supported by our in vitro experiments, showing that iNOS inhibition in lung epithelium does not significantly alter the response of these cells to injury, in terms of metabolic activity, proliferation and apoptosis. 

We previously demonstrated that iNOS inhibition can prevent and reverse PH and emphysema in mice, both after chronic smoke exposure or intratracheal elastase instillation, and that emphysema development was not dependent on iNOS activity in bone-marrow-derived cells [10,11,13]. Thus, in this study we aimed to identify the specific lung-cell type responsible for iNOS-dependent hindrance of alveolar regeneration in emphysematous lungs. Currently, available treatment options for COPD cannot cure the disease, nor can they stop disease progression. Only an alleviation of symptoms and reduction of future risk of exacerbations is hitherto possible. Interestingly, global inhibition of iNOS promoted lung repair and reverse remodeling of the pulmonary vasculature in two preclinical models of COPD, and might represent a novel treatment option, if transferrable to the human situation. However, for the transfer of such preclinical data to human COPD, delineation of the exact signaling mechanism of lung regeneration upon iNOS inhibition and the identification of responsible cell type(s) is necessary, as they could have a large impact on the therapeutic strategy (e.g., preferred route of administration, duration of treatment). In this regard, it has been suggested that failure of iNOS inhibition strategies in states such as sepsis and pain, might have been in part caused by insufficient understanding of iNOS’ complex functions and the dual modalities of iNOS and NO in a disease state (i.e., concentration-dependent protective versus harmful effects) [19]. Of note, several clinical trials showed good tolerability of iNOS inhibitors [20,21,22,23], and recent literature suggests that adult lung regeneration (after pneumonectomy) is possible in humans [24]. Hence, understanding the exact mechanism by which iNOS inhibition in emphysematous lungs elicits their regeneration and repair and identification of the responsible cell type is an important step toward the transfer of an iNOS inhibition strategy to a clinical trial and potential treatment of emphysema in humans.

Here, we hypothesized that deregulation of iNOS in AECII interferes with their role in lung regeneration, and that consequently, iNOS knockout in this cell type can promote the repair of alveolar epithelium in emphysematous lungs. Among numerous non-bone-marrow lung cell types, we chose AECII as the most likely contributor to the iNOS-dependent impediment of lung repair, based on their role in adult lung regeneration on the one hand and in the pathogenesis of COPD on the other [25,26]. Specifically, the proliferation of AECII and their transdifferentiation into AECI play a critical role in the restoration of normal alveolar architecture after injury [27], while elevated apoptosis of these cells has been linked to emphysema pathology [28]. Importantly, the previously mentioned study from our group demonstrated that the strong nitrating agent peroxynitrite, known to arise from the reaction of superoxide with iNOS-derived NO, could induce apoptosis of AECII [10], and Boyer et al. confirmed that emphysematous murine lungs are characterized with the marked upregulation of iNOS and the accumulation of nitrated proteins, predominantly in AECII [14]. However, to the best of our knowledge, this is the first study to attempt long-term knockout of iNOS, specifically in the AECII, as a therapeutic option for severe emphysema and pulmonary hypertension. 

To test whether iNOS deletion in AECII can lead to regeneration of the alveolar epithelium in emphysematous lungs, we employed the *CCSP rtTA2^s^-M2 LC1* driver line, which enables tight temporal and spatial control of expression of luciferase and Cre recombinase by doxycycline, and was previously reported to efficiently target AECII in the adult lung [18]. By crossing this line with *iNos^flox/flox^* mice, we achieved inducible, doxycycline-controlled deletion of iNOS, predominantly in AECII. Importantly, a previous study from our group demonstrated that, although doxycycline inhibits MMPs and has anti-inflammatory properties, treatment with this antibiotic alone does not influence the reversal of emphysema and PH in mice [11]. After the establishment of emphysema in the elastase mouse model, we induced iNOS knockout in AECII and gave an additional 12 weeks for the potential regeneration to occur. We chose this time point because previously Boyer et al. demonstrated that global iNOS knockout did not affect the emphysema severity at shorter time points, while the study from our group showed that long-term (12 weeks) treatment with iNOS inhibitor L-NIL could ameliorate emphysema in the same animal model [11,14].

Surprisingly, AECII-specific knockout of iNOS did not promote regeneration of the alveolar epithelium in the elastase mouse model. These data were corroborated by in vitro findings, where inhibition of this enzyme could not significantly alter the response of AECII and MLE-12 cells upon injury. In addition to AECII-specific knockout, the fact that in injured lungs AECII proliferate and give rise to AECI [25] implies that in our elastase-treated knockouts at least a portion of AECI was also iNOS-deficient. Although our study did not determine exact portions of respective cell types with successful iNOS knockout, the fact that we failed to detect even mild beneficial effects of iNOS knockout implies that a lung compartment other than the alveolar epithelium might be a primary location of detrimental iNOS activity in the context of lung repair. Taken together with previous findings from our laboratory, our current results suggest that another lung cell type, such as pulmonary vascular cells or fibroblasts, may play a critical role in emphysema reversal upon iNOS inhibition. Strong induction of iNOS expression in the pulmonary vasculature of cigarette smoke-exposed mice and COPD patients detected in our previous studies, supports this notion [10,13]. Alternatively, the idea should be explored that lung stem cells other than AECII are more important players in the regeneration of emphysematous lungs. Recently, evidence was provided that even AECI can convert to AECII to support lung regrowth after a pneumonectomy [29]. Moreover, a unique cell population at the bronchioalveolar-duct junctions with stem-cell properties has been identified, and shown to expand upon alveolar injury and differentiate into AECI and AECII [30]. Nevertheless, even if the knockout of AECII-derived iNOS cannot ameliorate already established severe pathology, the question remains whether iNOS upregulation in alveolar epithelium contributes to the development of emphysema. To test this option, however, another animal model, such as exposure to cigarette smoke, would be more appropriate, as the mechanism of emphysema development in the elastase model relies on the fast enzymatic destruction of the alveolar architecture, and thus does not necessarily mimic the gradual and complex molecular changes driving emphysema in humans [31].

Similarly, conclusions regarding the potential role of AECII-derived iNOS in reverse remodeling of the pulmonary vasculature have to be drawn with caution, as the scarce data available in the literature suggest that the pulmonary vascular pathology in the elastase model arises from the loss of capillary bed and hypoxemia [9,32,33,34]. Nevertheless, elastolysis and extracellular matrix remodeling have indeed been suggested as important and early events in PH development [35,36,37]. Similarly, inflammation, another important contributor to pulmonary vascular remodeling, is present in this model as well [31]. In addition, the previously reported beneficial effect of iNOS inhibition on the pulmonary vasculature in this model argues in favor of its similarity to the situation observed in cigarette smoke-exposed mice and human smokers. Finally, the ability of the iNOS inhibitor L-NIL to ameliorate elastase-induced PH postulates that iNOS knockout would be effective as well if carried out in the respective underlying cell type. We have previously shown that iNOS deregulation in myeloid cells, specifically macrophages, plays an important role in pulmonary vascular pathology in COPD. However, it remained unclear whether the knockout of myeloid cell-derived iNOS is also sufficient to promote the reverse remodeling of pulmonary vasculature. In this regard, it is important to consider the fact that the synergistic/simultaneous effects of iNOS in different cell types could drive lung regeneration. Nevertheless, iNOS knockout in AECII did not produce even a slight, partial improvement in any physiological or histological parameter used for the evaluation of pulmonary vasculature and the right ventricle. This failure to activate reverse remodeling of the pulmonary vasculature with AECII-specific iNOS knockout prompts us to conclude that (an)other lung cell-type(s), either bone-marrow-derived or of another origin, mediate(s) the beneficial effects of iNOS inhibition on remodeled pulmonary vessels in preclinical models of COPD.

As our study focused on the long-term effects of cell-specific iNOS knockout, it does not provide a detailed description of deregulated signaling pathways and physiological processes in early time-points after the injury with elastase, but rather gives insight into the chronic disease state. This is well exemplified by the fact that signaling pathways, such as those governing apoptosis and proliferation, are not deregulated in animals challenged with elastase, contrary to what one would expect after severe lung injury. However, elevated levels of MMP-8 could still be observed in elastase-treated mice, suggesting its possible involvement in the maintenance (or even propagation) of the emphysema pathology. Curiously, among the rare molecular changes we were able to detect in the lungs of elastase-treated mice 12 weeks after the establishment of the pathological phenotype, was a decrease in cytochrome c content that was more prominent in the control than in the AECII-specific iNOS knockout animals. As this change was not accompanied by alterations in caspase 3 activation, we conclude that it is limited to the mitochondrial compartment. The possibility remains that a decrease in the cytochrome c level reflects a reduction in the number of mitochondria, or influences the function of the mitochondrial respiratory chain, but the functional effects and overall significance of this finding remain to be investigated.

Another interesting discovery, warranting further investigation, relates to the in vitro effects of BALF on the apoptosis of the MLE-12 cell line. Namely, the decrease of apoptosis observed in cells treated with BALF from elastase-challenged lungs implies the existence of damage-control mechanisms in these lungs, mediated through as yet unknown soluble factors. Identification of such factors and careful delineation of the signaling pathways they activate might help design novel therapeutic concepts for COPD.

## 5. Conclusions

To the best of our knowledge, this study is the first to demonstrate that knockout of *iNos* in the AECII cannot promote the repair of emphysematous lungs or stimulate the reversal of PH in the elastase mouse model during a 12-week observation period. Previously reported beneficial effects of iNOS inhibition on emphysema and PH reversal in preclinical models of COPD are thus likely mediated by (an)other lung cell-type(s), such as fibroblasts or (other) pulmonary vascular cells. Identification of the exact cell type could help design an optimal treatment strategy for the possible future use of iNOS inhibitors in COPD patients, if our data from mouse models are transferable to the human situation.

## Figures and Tables

**Figure 1 cells-12-00125-f001:**
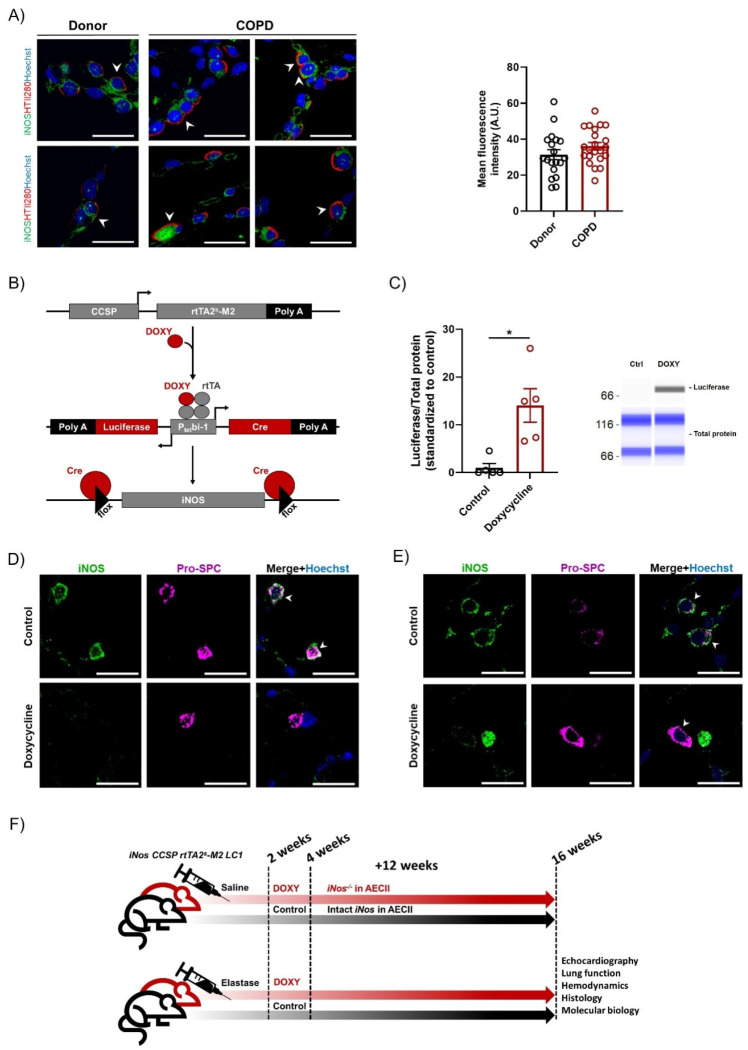
iNOS is successfully deleted in AECII using *iNos CCSP rtTA2^s^-M2 LC1* mouse line. (**A**) Representative images of human tissue sections from donor and COPD lungs, co-stained against iNOS (green) and AECII apical membrane protein HTII280 (red), and counter-stained with Hoechst. Fluorescence intensity of the iNOS staining in AECII (HTII280-positive cells) was quantified. Arrows indicate double-positive cells. Scale bar = 20 µm. For the signal intensity quantification, every point on the graph represents quantification of the mean fluorescence intensity in one AECII. No fewer than 4 images per group were quantified. (**B**) Schematic representation of the genetic constructs allowing for club-cell secretory protein (CCSP)-reverse tetracycline-dependent transactivator (*rtTA2^s^-M2*)–mediated, doxycycline-controlled luciferase (Luc) and Cre expression and iNOS deletion in mice. (**C**) Automated Western-blot analysis (*n* = 5) of luciferase expression in AECII isolated from mice fed with either normal (control, ctrl) or doxycycline-containing food. (**D**,**E**) Representative images of lung sections from control and AECII-specific iNOS knockout mice, immediately (**D**) and 12 weeks (**E**) after the termination of doxycycline feeding for cell type-specific iNOS knockout induction. The samples were co-stained against pro-SPC (magenta) and iNOS (green) and counterstained with Hoechst (blue). Scale bar = 20µm. Arrow: AECII (pro-SPC-positive) cells. (**F**) Schematic representation of experimental design used to assess the effect of iNOS expression in AECII on the regeneration of alveolar epithelium and the reversal of pulmonary hypertension in an elastase mouse model. DOXY: Doxycycline. A.U.: Arbitrary units. Graphs show mean ± SEM. * *p* < 0.05. An independent *t*-test was used.

**Figure 2 cells-12-00125-f002:**
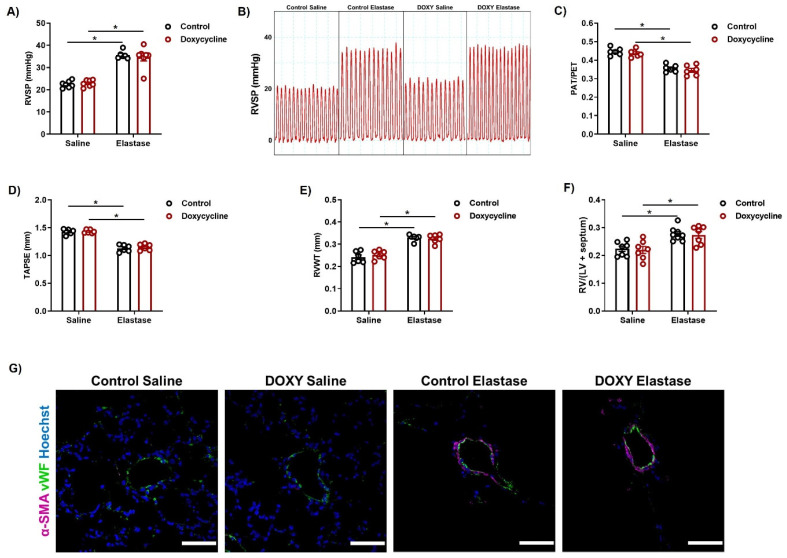
iNOS knockout in AECII cells does not promote reverse remodeling of the pulmonary vasculature upon elastase instillation. Two weeks after intratracheal instillation of saline or elastase, *iNos CCSP rtTA2^s^-M2 LC1* mice were fed with either standard (control) or doxycycline-containing (Doxycycline, DOXY) chow for additional two weeks. Twelve weeks later, right heart function and structure as well as the vascular structure were investigated. (**A**,**B**) Right ventricular systolic pressure (RVSP) (*n* = 6–7 per group). (**C**–**E**) Echocardiographic assessment of right ventricular function and dimensions (*n* = 6 per group), depicted as (**C**) the ratio (PAT/PET) of pulmonary acceleration time (PAT) and pulmonary ejection time (PET), (**D**) tricuspid annular plane systolic excursion (TAPSE), and (**E**) right ventricular wall thickness (RVWT). (**F**) Changes in right ventricular weight shown as the ratio of the right ventricular (RV) and the left ventricular plus septum (LV + septum) mass (*n* = 7–8). LV mass was not significantly altered. Graphs show mean ± SEM. * *p* < 0.05. Two-way ANOVA (with Tukey’s multiple comparison post-hoc test) was used. (**G**) Representative images of pulmonary vessels in lung sections from control and doxycycline-treated mice instilled either with saline or elastase, co-stained against α-smooth muscle actin (α-SMA, magenta), von Willebrand factor (vWF, green), and counterstained with Hoechst (blue). Scale bar = 50 µm.

**Figure 3 cells-12-00125-f003:**
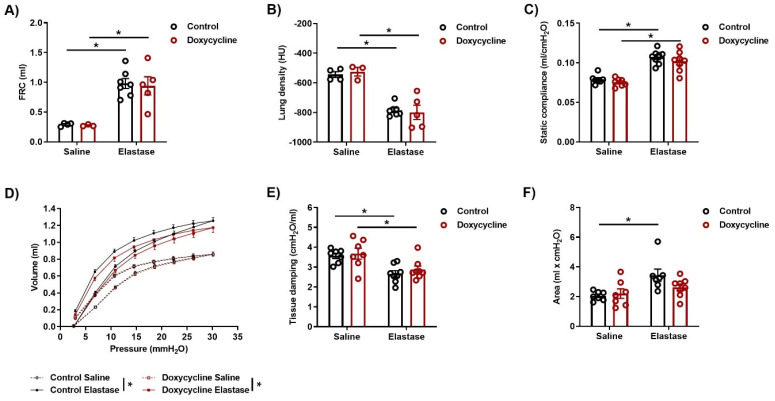
Deletion of iNOS in AECII cannot ameliorate elastase-induced changes in lung function. Two weeks after intratracheal instillation of saline or elastase, *iNos CCSP rtTA2^s^-M2 LC1* mice were fed with either standard (control) or doxycycline-containing (Doxycycline) chow for additional two weeks. Twelve weeks later, micro-computed tomography (µCT) and lung function measurements were performed. (**A**,**B**) µCT-based assessment (*n* = 3–7) of (**A**) functional residual capacity (FRC) and (**B**) lung density. HU: Hounsfield units. (**C**–**F**) Lung function (n = 7–8) presented as (**C**) static compliance, (**D**) respiratory pressure-volume (P-V) loops, (**E**) tissue damping, (**F**) area enclosed by the P-V loop. Graphs show mean ± SEM. * *p* < 0.05. Two-way ANOVA (with Tukey’s multiple comparison post-hoc test) was used.

**Figure 4 cells-12-00125-f004:**
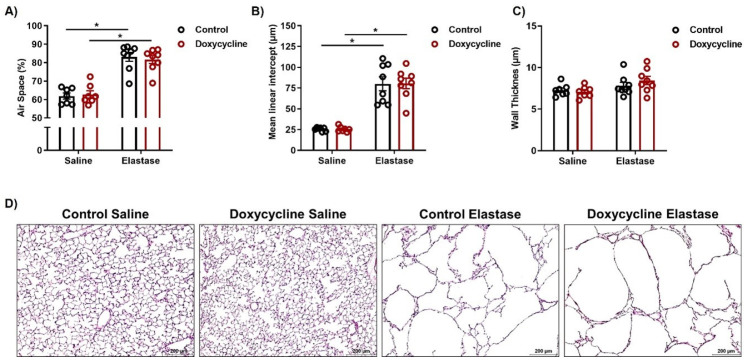
iNOS knockout in AECII cells does not influence regeneration of alveolar epithelium in mice with severe emphysema. (**A**–**C**) Alveolar morphometry (*n* = 7–8) showing (**A**) the percentage of airspace, (**B**) septal wall thickness and (**C**) mean linear intercept in saline- or elastase-treated *iNos CCSP rtTA2^s^-M2 LC1* mice fed either with normal (control) or doxycycline-containing (Doxycycline) chow, after the 12-week follow-up observation period. Graphs show mean ± SEM. * *p* < 0.05. Two-way ANOVA (with Tukey´s multiple comparison post-hoc test) was used. (**D**) Representative images of lung sections from control and doxycycline-fed mice, 12 weeks after elastase instillation, stained with hematoxylin-eosin (H&E). Scale bar = 200 µm.

**Figure 5 cells-12-00125-f005:**
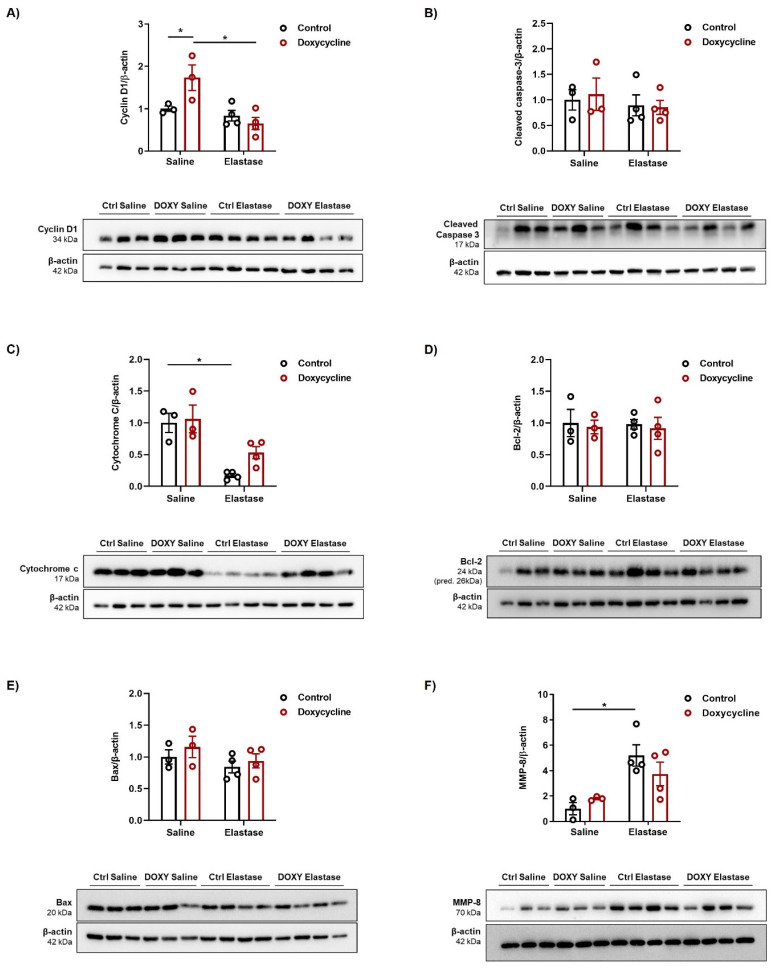
iNOS knockout in AECII cells influences Cytochrome c expression in elastase-treated mice. Western-blot analysis (*n* = 3–4) of (**A**) Cyclin D1, (**B**) Cleaved caspase 3, (**C**) Cytochrome c, (**D**) Bcl-2 (**E**) Bax and (**F**) MMP-8 in lung homogenates of saline- or elastase-treated *iNos CCSP rtTA2^s^-M2 LC1* mice fed either with normal (control, Ctrl) or doxycycline (Doxycycline, DOXY)-containing chow, after the 12-week follow-up period. Data are given as the ratio between the protein of interest and β-actin, standardized to saline control. Graphs show mean ± SEM. * *p* < 0.05. Two-way ANOVA (with Tukey’s multiple comparisons post-hoc test) was used.

**Figure 6 cells-12-00125-f006:**
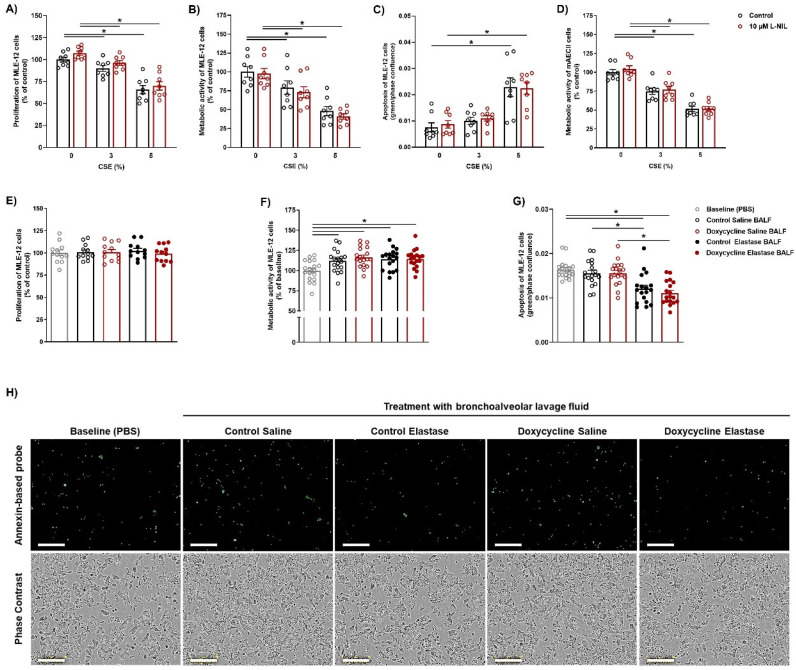
Bronchoalveolar lavage fluid from elastase-treated mice decreases apoptosis of mouse lung epithelial (MLE-12) cell-line in vitro. MLE-12 cells were treated with cigarette smoke extract (CSE) for 6 h, and then left for an additional 16 h in the presence or absence of L-NIL. (**A**) Proliferation of MLE-12 cells during the 16 h observation period assessed using the BrdU incorporation assay and standardized to control (*n* = 8). (**B**) Metabolic activity of MLE-12 cells assessed after the 16 h observation period using the alamarBlue assay (*n* = 8). (**C**) Apoptosis of MLE-12 cells after the 16 h observation period given as the confluence of cells labeled with Annexin XII-based polarity-sensitive probe, standardized to total cell confluence (*n* = 8). (**D**) Metabolic activity of primary murine alveolar epithelial type 2 cells (AECII) assessed using alamarBlue assay during the 16 h observation period (*n* = 8). (**E**–**H**) MLE-12 cells were treated with 10% of either PBS (control) or bronchoalveolar lavage fluid (BALF) from saline- or elastase-treated *iNos CCSP rtTA2^s^-M2 LC1* mice, fed either with normal (control) or doxycycline-containing chow as above, taken after the 12-week observation period. (**E**) Proliferation of MLE-12 cells during 24 h of treatment, assessed using the BrdU incorporation assay and standardized to control (*n* = 12). (**F**) Metabolic activity of MLE-12 cells after 24 h of treatment, measured using the alamarBlue assay and standardized to control (*n* = 18). Graphs show mean ± SEM. * *p* < 0.05. One-way ANOVA (with Tukey’s multiple comparison post-hoc test) was used for statistical analysis. (**G**) Apoptosis of MLE-12 cells during a 48 h long treatment given as the confluence of cells labeled with Annexin XII-based polarity-sensitive probe, standardized to total cell confluence (*n* = 18). Graphs show mean ± SEM. * *p* < 0.05. Two-way ANOVA (with Tukey’s multiple comparison post-hoc test) was used for statistical analysis. (**H**) Representative photos from apoptosis-kinetic assay. Green: Annexin XII-based polarity-sensitive probe. Scale bar = 300 µm.

## Data Availability

The raw data can be provided by the corresponding author upon reasonable request.

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
