# Peer review of "iNOS Deletion in Alveolar Epithelium Cannot Reverse the Elastase-Induced Emphysema in Mice"

_cells, 2022, doi:10.3390/cells12010125_

Round 1

Reviewer 1 Report

In this paper, Norbert Weissmann and collaborators report the lack of effect of iNOS deletion in type II alveolar pneumocytes (AECII) on elastase-induced emphysema formation in mice. The authors employed conditional knockdown enabling tight temporal and spatial control of expression of luciferase and Cre re- 402 combinase by doxycycline efficiently targeting AECII cells, and inducible, doxycycline-controlled deletion of iNOS predominantly in these cells.
Hence, the authors report a “negative result”, which is quite unexpected considering previous reports, that iNOS may play a significant role in nicotine- and elastase – induced emphysema formation. In general, negative reports bear the risk that experimental faults or flaws might have veiled the demonstration of an existing effect. However, in this specific paper, I cannot recognize a single flaw. All experiments were done with the appropriate controls, the paper is very well written, and the experimental design looks very reasonable to me.

In summary, I believe that this is a very important finding that deserves publication. The conclusion that iNOS in cell types different from AECII drives emphysema formation, aims at combining previous positive findings with this negative one. Alternatively, it appears to me that iNOS may play no role at all in elastase-induced emphysema, and previously reported pharmacological evidence (Br J Pharmacol 2021, 178, 152-171) was just a non-specific effect of a drug.

Reviewer 2 Report

The present study explored the potential role played by iNOS in the reversion of established pulmonary emphysema. The study is mainly based on a 12 weeks recovery of a mouse model with a knock-out of iNOS protein in epithelial alveolar cells exposed to elastase, a well-known model of severe emphysema. This study opens interesting hypotheses to better understand the role played by iNOS in this pathology but none of them have been truly explored.

Major comments:

1)            In figure 1A, please add a quantification of the iNOS signal intensity to compare donor and COPD conditions. It is not clear that COPD presents a higher expression of iNOS in AECII despite its known increase in the lung tissue of these patients. This is in line with the absence of effect observed in the ko’s mice and with your conclusions.

2)            Some western blots of figure 5 show important variations of the loading control (i.e. actin) and the membrane of the cleaved caspase 3 (fig 5B) is probably torn. Please find better images.

3)            The link between iNOS KO mice and the cell lines exposed to CSE is hard to understand. Adding a paragraph in the introduction may help to merge these two different experiments in one study.

4)            Figure 6G shows that apoptosis is decreased in MLE-12 cells exposed to BALF from mice treated with doxycyclin independently of elastase treatment whereas the authors suggest the opposite in lines 349-352. Please clarify this point. Furthermore, I think that characterization of the BALF (i.e. number of cells, types of cells, and protein concentration) could be very interesting to fully interpret the results of figure 6.

Minor comments

1)            Contrast of the annexin pictures in Figure 6H is very low, making the figure hard to interpret.

2)            Some figure legends are in bold and some are not, please normalize them.

3)            In material and methods it could be interesting to add a reference to figure 1F to help the reader to understand the model of the study.

4)            Some abbreviations are not explained (e.g. CSE) please check carefully.

5)            Line 215 : the size of characters has to be normalized

Reviewer 3 Report

The manuscript by Gredic et al. titled "iNOS deletion in alveolar epithelium cannot reverse the elastase-induced emphysema in mice" report the absence of a role of iNOS, expressed in alveolar epithelial cells, in the reversal of emphysema. The manuscript is reporting negative findings of the hypothesis of the involvement of iNOS in emphysema. This condition known for causing shortness of breath and characterized by the destruction of the alveoli wall leading to the formation of larger ineffective air sacs. The primary cause of emphysema is the degradation of the lung tissue by elastase, an enzyme released by inflammatory activated macrophages and neutrophils. The authors attempted to determine the source of the initial signal responsible of development of emphysema and their findings lead them to conclude that alveolar epithelial cells were not responsible and maybe some other still unknow respiratory cells were responsible for the initiation of emphysema.

There was one single positive result in the experiment using BAL from animals with emphysema. The BAL contained an unknown compound that inhibited the effect of elastase; however, no attempt was made to determine the nature of this compound. One may say the aim of this research was not to determine what reverses emphysema but what initiates it.
